# Perception and Deception: Human Beauty and the Brain

**DOI:** 10.3390/bs9040034

**Published:** 2019-03-29

**Authors:** Daniel B. Yarosh

**Affiliations:** Retired SVP Basic Science Research, Estée Lauder Co., Merrick, NY 11566, USA; dyarosh@danyarosh.com

**Keywords:** facial beauty, attractiveness, evolutionary biology, costly signals, cosmetics, deception, self-signaling

## Abstract

Human physical characteristics and their perception by the brain are under pressure by natural selection to optimize reproductive success. Men and women have different strategies to appear attractive and have different interests in identifying beauty in people. Nevertheless, men and women from all cultures agree on who is and who is not attractive, and throughout the world attractive people show greater acquisition of resources and greater reproductive success than others. The brain employs at least three modules, composed of interconnected brain regions, to judge facial attractiveness: one for identification, one for interpretation and one for valuing. Key elements that go into the judgment are age and health, as well as symmetry, averageness, face and body proportions, facial color and texture. These elements are all Costly Signals of reproductive fitness because they are difficult to fake. However, people deceive others using tricks such as coloring hair, cosmetics and clothing styles, while at the same time they also focus on detecting fakes. People may also deceive themselves, especially about their own attractiveness, and use self-signally actions to demonstrate to themselves their own true value. The neuroscience of beauty is best understood by considering the evolutionary pressures to maximize reproductive fitness.

## 1. Introduction

Human nature includes a desire to be attractive, and historically much of the fine arts are depictions of human beauty. Much time, money and emotional energy are spent in improving our appearance to reach a goal of beauty. People feel better about themselves when they think they are attractive to others. We devote portions of our brains to evaluating characteristics of attractiveness that are remarkably similar among cultures. Our bodies are shaped not only for function but also to match the image of attractiveness to others.

The simple answer that “beauty is for attracting mates” is no longer sufficient to explain the wealth of data on human preferences for beauty. Attractiveness is part of our status ranking among our same-sex peers, and we actively deceive others and ourselves about our personal appearance.

This review is crafted to place the study of personal appearance and beauty in the context of evolutionary biology. This theoretical framework best explains the quirkiness, universality and unexpected behaviors of people striving to be attractive and seeking out beautiful people.

## 2. The Evolutionary Biology of Beauty

The principle of evolutionary biology is that when there is genetic variation within a population of a characteristic that improves the individual’s chance of survival and reproduction (sending its genes into several succeeding generations), that characteristic with the best improvement will be naturally selected over other forms and becomes more common within the population. Eventually that characteristic phenotype will become nearly universal, and the genetic variant producing the favored phenotype will become “fixed” in the population.

Some phenotypes improve an individual’s acquisition of food, such as running stamina or manual dexterity. However, those phenotypes that directly improve the chances of reproduction, such as attracting mates and obtaining help in raising children, are under even stronger selection pressure, since they directly influence the frequency with which those genes are passed to the next generation.

### 2.1. Reproductive Strategy

Men and women have different strategies for reproductive success that were honed during tens of thousands of prehistoric years. Women seek men for partners who will contribute material resources as well as good genes to their children, while men seek one or more female partners with good genes, some of whom they may provide with resources. The strategy of each sex includes advertising to potential mates, and competing members of the same sex, to demonstrate that he or she is valuable (reviewed in [1]). The display of these traits is called “attractiveness” or “beauty”.

### 2.2. Universality of Attractiveness Judgements

Assessments of attractiveness are surprisingly similar between men and women and among groups of people. A meta-analysis, covering 919 studies and over 15,000 observers, reported that people agree, both within cultures and across cultures, who is attractive and who is not [2]. Men and women as well as people of all ages agree on who is attractive. This strongly suggests that judgments of physical attractiveness are hard-wired in human genetics, likely fixed at an early stage in our evolution. These assessment tools are available at a remarkably early stage of human development. Six-month-old infants gazed longer at faces judged by adults as attractive and spent less time looking at faces that were judged as not attractive [3].

### 2.3. Attractive People Succeed

Judgments of attractiveness have real consequences because they are cues of a person’s health and fitness, which indicate the ability to donate good genes and successfully raise children. Attractiveness is the most important predictor of who gets the preferred choice in mates [4]. In fact, in the modern world, physical attractiveness is significantly associated with reproductive success [5]. A woman who chooses a male partner who contributes not only good genetic material but also provides resources will on average be more successful than a woman without such support [6].

This means that attractiveness *and the ability to accurately detect attractiveness* are under evolutionary selective pressure. Therefore, it is not surprising that the brain has developed specialized systems to accurately assess attractiveness characteristics, such as age, health and reproductive potential.

## 3. The Neuroscience of Facial Recognition

### 3.1. Brain Loci

The most extensive research on the brain regions used in assessing beauty has been reported for facial recognition [7] and less research has been reported on body judgments [8]. Brain loci used to judge the beauty of faces are distinct in distribution and activation intensity from those used to assess the beauty of non-facial visual art [9], reflecting the evolutionary salience of facial beauty. While a few loci have been linked together to suggest a pathway for the evaluation of beauty, this is not to suggest that this is the only way the brain makes these judgments and, under special conditions, the plasticity of the brain may invoke other regions to participate in reaching assessments.

The brain uses at least three modules, or cognitive domains, in deciding the value of attractiveness. The occipital and temporal regions of the cortex are used first to process face views [10]. The inferior occipital gyri (IOG) perceives facial features and passes the information to the fusiform face area (FFA) of the fusiform gyrus (FG) for facial recognition [11]. The FFA recognizes and processes the location of facial features (especially the eyes, nose, and mouth) and their spacing [12]. People have distinct eye movement patterns (scan path routines) when they judge unfamiliar faces [13], and they simultaneously engage the FFA region during this routine [14]. Damage to the FFA causes prosopagnosia, a condition in which patients are unable to recognize faces by sight or accurately judge facial attractiveness, although they can recognize the same people by voice [9,15]. The FG very quickly responds more strongly to attractive faces than unattractive ones [16], suggesting that the ease of recognition of attractive features occurs perhaps even before the rest of the brain is included in the evaluation.

The IOG connects to the second module, including the superior temporal sulcus (STS) for interpretation of facial movement, such as eye gaze, lip movement and facial expressions [8]. The FFA and IOG then interact with other brain regions, such as the occipital face area (OFA) and the ventral anterior temporal lobes (vATLs) for feature abstraction and assessment [17], and the amygdala, insula and limbic system for the emotional content of facial expressions and movement [8].

Information from the STS is also passed to the third module, the orbitofrontal cortex (OFC), including the nucleus accumbens, for making judgments of beauty and producing the neurological rewards (dopamine and other neurotransmitters) for finding it [18]. The OFC responds with greater activity to attractive versus unattractive faces [6]. When men were shown faces of beautiful women while their brains were scanned by fMRI, the attractive faces specifically activated the nucleus accumbens in the caudate region of the brain, when compared to viewing average faces [19]. Transcranial stimulation of the dorsolateral prefrontal cortex (dlPFC) increased the perceived attractiveness of faces but did not affect other facial judgments such as age [20]. These studies suggest that the value but not the features of the face are decided in these third module cortical regions.

Human bodies, both self and others, are selectively perceived in the temporal lobes by the extrastriate body area (EBA) and the fusiform body area (FBA), whether they are full body representations, stick figures or silhouettes [21]. The OFC, particularly the nucleus accumbens and anterior cingulate cortex, are then used in judging the beauty of nude bodies [22]. Similar regions of the brain are used in evaluating sculptures and similarly posed real human bodies [23].

### 3.2. Gender-Specific Brain Activation

Male and female brains activate differently while evaluating appearance and beauty, consistent with their differences in reproductive strategy. For heterosexuals, opposite-sex faces stimulate assessment and reward brain systems, such as the amygdala, cingulate and insular cortices, more than same-sex faces, signifying they hold greater salience [6]. Both heterosexual men and women favor viewing attractive faces, but men willingly expend more effort to view beautiful women’s than men’s faces, while women spend less energy, and equivalent amounts, to view both beautiful men’s and women’s faces [24]. Men show slower response times to beautiful faces than women, evidencing greater cognitive load while processing attractive faces [25]. Consistent with this, brain imaging studies show that the ventromedial prefrontal cortex (vmPFC) of male subjects is more sensitive to physical attributes, such as the youthfulness and gender of faces, than female subjects [26].

## 4. Appearance and Beauty Judgments

### 4.1. Gender Differences in Attractiveness Features and Perception

The sex hormones, testosterone in men and estrogen in women, largely drive the body and facial features that define attractiveness, and also reshape the brain to detect and value these features. The onset of puberty ramps up hormone levels and reshapes the male and female bodies. Men increase their shoulder to waist ratio, their beards grow, and their jawlines become more pronounced. For women, breasts develop, the hips to waist ratio increases, and their jawlines and facial features become softer. Several regions of the brain express either estrogen/progesterone receptors or androgen receptors, and brain structure and responses are, therefore, on different developmental trajectories in men and women beginning in puberty [6].

Women are so attuned to the facial features of men that simply by looking at their photographs they can correctly rank order a group of men based on their saliva testosterone level [27]. Interestingly, while a woman tends to prefer a man with high testosterone for an affair, she prefers a little less testosterone for a long-term mate, and her parents (who might have to help take care of any babies if the man leaves) tend to prefer even a little less testosterone [28]. Exaggerating the masculinity of men’s pictures actually makes them less attractive [26].

On the other hand, estrogen monotonically drives female beauty [29]. Panels of men and women were shown women’s faces that were morphed to exaggerate feminized features, and 95% of men and women decided that the feminization of women’s faces made them more attractive. The same result was found for faces of European, African and Asian descent [30]. In another study, estrogen was measured in women over the course of their monthly cycles. Both men and other women rated their attractiveness. Women with higher estrogen levels had higher ratings of femininity, attractiveness and health. Interestingly, when the women wore color cosmetics, the correlation disappeared, suggesting that makeup literally “makes up” for lower estrogen levels [31].

### 4.2. Age Perception

Youth is a major component of facial attractiveness [32] and underlies most of the specific characteristics people look for in judging attractiveness. Older faces are judged as less attractive, less likeable, less distinctive, and less energetic [33]. The *appearance* of aging past the prime of life is an assault on self-esteem and confidence [34].

In particular, age is used, along with other skin and body signs, to assess standing in the community, desirability as a partner, and reproductive potential [35,36]. Traditionally, men more than women tend to accumulate resources with age (which makes them more attractive partners), while women more than men tend to lose fertility with age. As a result, the sharp decline in attractiveness with women’s age after menopause is largely driven by male perception, while the perception of increased power of men with age is predominantly due to female opinions [37].

People are exquisitely sensitive to the age of others and are excellent judges of each other’s age, with a correlation coefficient of perceived to actual age of 0.95 [38]. Just by viewing a swatch of skin people are able to correctly judge age, with a correlation coefficient of more than 0.60 [39]. The most important factors in judging age from facial images are the size of the eyes and the lips, and the evenness of skin tone, regardless of what that tone might be [34].

### 4.3. Health Perception

People use specific cues from the appearance of others to make judgments about that person’s health, including the history as well as the current state of health. Assessments of health often overlap assessments of age in determining beauty, such as in the case of the sclera, or white part of the eyes. Sclera become darker and colored with age or poor health, and the whiteness of sclera are strongly correlated with the perception of youth, health and attractiveness [40].

### 4.4. Symmetry

Throughout the animal kingdom, and certainly among people, body symmetry is a strong signal of past and present health. Bilateral symmetry is a sign of the absence of congenital or developmental defect, malnutrition or parasitic infection, all of which are common maladies in subsistence living [41]. Although minor variations are often of no functional consequence, they do have dramatic impact on the perception of beauty [42]. The absence of a history of pathology is a good sign of reproductive fitness and the preference for symmetry is culturally universal [43], suggesting that it is hard-wired into brain judgments by natural selection and not derived from culture.

Women prefer men with symmetrical faces and can select symmetrical men from their scent [6]. Unfortunately for cologne manufacturers, the scent of androstenone is an unpleasant under-arm smell. Her preference for symmetry is even heightened during a woman’s fertile phase of her monthly cycle [44], an effect found for several female preferences that is called *ovulatory shift*.

### 4.5. Average Features

Sir Francis Galton, a cousin of Charles Darwin and the inventor of fingerprinting, was studying the faces of criminals to identify diagnostic facial features and was the first to report that the average face prepared by composites of criminal faces is more attractive than individual faces [45]. The preference for the average is found among all cultures [46] and strengthens as children develop after 5 years of age [47]. Interestingly, male preference for women’s faces is correlated to facial averageness, but women’s self-perceived attractiveness is not correlated with their averageness [48].

There may be several reasons for the preference for average facial features. One trivial explanation is that the preparation of composites tends to smooth out asymmetrical or uneven features of individuals. A second is that the very nature of cultural learning favors the most common feature or practice [49]. A third is that, assuming facial features are under genetic control and are adaptive, natural selection will favor a fitness peak that we perceive as average compared to the less fit facial forms. Finally, the beauty of average may lie in the fact that it is most expected and imposes the least cognitive load to recognize and interpret [4].

Despite the preference for average, exaggerating some key facial features actually improved the attractiveness of faces [26], meaning that average is attractive, but unusually endowed faces may be more attractive. This phenomenon (found throughout the animal kingdom) is termed *signal shift*: a preference for an elemental characteristic, and a heightened preference for the extreme form [50]. The signal shift response may identify a simple characteristic the brain overweighs in valuing faces. There is a limit to the attractiveness of exaggerated features, and an extreme form that is outside the range of normal experience causes extra work for the brain, which then considers the form weird and ugly.

### 4.6. Face Proportions

As social creatures, humans read other people’s intentions and emotions in their faces and adjust their behavior accordingly. People also view the face as an important determinant of attractiveness, which is a signal of reproductive fitness. As discussed previously, people of both sexes and nearly all cultures agree on which faces are attractive and which are not. The features people find attractive are shaped in part by sex hormones, including stronger or softer jaw and larger or smaller eye shape [51]. However, the cognitive processes that determine attractiveness are not always accessible to consciousness. Composites were prepared from a group of college students, one composite from those judged most attractive and another from those judged least attractive [26]. People who view the two composites side by side can agree on which is more attractive, but it is difficult to put into words which features lead to the decision.

The eyes are a specific target of the human face for social interaction and beauty assessment. The brain uses a special region, the superior temporal sulcus, for the job of following eye movements in others and determining the direction of their view [52]. This region develops early, and neonates learn within months to follow their mothers’ gaze [53]. Eye recognition is wired directly into the most fundamental emotional processing unit in the brain—the amygdala [54].

By contrast, the prominence of the ears and nose are not signals of beauty, because the length of both the ears [55] and nose [56] relative to the rest of the face continue to increase with age.

### 4.7. Body Proportions

Body shape is also a signal of reproductive fitness. People universally have a preference for shape as expressed in the ratio of waist to hips for men of 0.9, and waist to hips for women of 0.7 [36]. In one study, men were shown pictures of naked women before and after surgery that improved their waist-to-hip ratio to be closer to 0.7 and found that approaching the ideal specifically activated the orbitofrontal cortex and the anterior cingulated cortex [57], regions that are also used in judging the attractiveness of faces. These preferences arise early, with children as young as 3-years-old selecting canonical body shapes over those with altered legs to trunk ratios [58].

The movement of other people is also of great interest to social humans. The extrastriate body area (EBA) of the occipitotemporal area is selectively activated in evaluating human bodies and movement [21] and such activation for heterosexuals is greater for opposite-sex bodies than same-sex bodies [59].

### 4.8. Foot Size

As women age and bear children, the size of their feet relative to their height increases [60,61]. A proportionately small foot is therefore a signal of youth and untapped reproductive potential. Not surprisingly, both men and women prefer small feet in women. In a series of studies, images of both men and women were altered to increase or decrease the size of the foot relative to height. Observers of the images, both men and women, preferred the natural proportion of foot to height in men over exaggerated smaller or larger feet. But they predominately judged the disproportionately smaller foot of women as more attractive than the natural proportion [62,63]. High heeled women’s shoes achieve the appearance of a smaller foot by raising the heel relative to the toe and shortening the distance from the heel to the toe in the footprint. The shoe generally just covers the toes and not the instep, further accentuating the appearance of a small step.

### 4.9. Facial Color and Wrinkles

Despite the variation in underlying skin tones, the homogeneity of skin color is correlated with increased attractiveness and appearance of healthiness in every culture examined [64,65]. People who view a cropped image of a cheek were able to accurately judge age, based on the homogeneity of skin tone [39]. In traditional Chinese medicine, skin color is used as a diagnostic tool for disease [66].

Among Caucasians, men perceive red tones in women’s faces as more attractive than less red faces because it is viewed as a sign of health. [67]. For these women, red facial coloration tracks their level of estradiol, and facial coloration may provide men with cues about fertility [68].

Facial color gradient is also important because as skin color gets darker with age, the color contrast between the hair, eyes and facial skin is reduced [69]. This is a consistent finding among many ethnic groups, including Caucasians, Chinese, Latin Americans and South Africans [70] Wrinkling increases with age and sun exposure and is a strong signal for judging age [71]. The discoloration of skin with age is interpreted as a loss of health, while wrinkling is perceived as a sign of age and loss of fertility [71].

## 5. Costly Signals and Deception

### 5.1. Costly Signals

The brain has evolved its focus on these features of attractiveness and beauty because not only are they reliable measures of health and reproductive fitness, but they are (or have been for most of our evolutionary history) difficult to fake. Such features that genuinely signal reproductive fitness are known as Costly Signals [72]. Costly Signals that are specific to one sex or the other are subject to strong sexual selection. One sex prefers a feature signaling reproductive fitness and chooses partners who have that feature. The next generation produces one sex that favor the Costly Signal and the other sex that displays it. This is called the Green Beard Effect, after the hypothetical example of two sets of genes that co-evolve, one set that produces a green beard in males and another set that prefers a green beard in females. This can lead to the run-away evolution of exaggerated features, such as extraordinarily large elk antlers or peacock feathers. The key element is that the Costly Signal must be biologically difficult to produce, or else fakers without the necessary reproductive fitness will display the feature and gain unwarranted mating opportunities.

Most if not all of the characteristics described here as signs of attractiveness and beauty are Costly Signals. They reflect healthy development, absence of disease, and display the level of sex hormones that reflect fertility. Human culture has added body ornaments and possessions to Costly Signals. These are additional signs of wealth and resource acquisition that also signal reproductive fitness, including decorative clothing and jewelry, lavish housing and luxury possessions.

### 5.2. Deception

Humans have also devised ways by which they deceive others as to their true reproductive fitness by faking Costly Signals.

#### 5.2.1. Makeup and Cosmetics

Women around the world apply makeup to alter their appearance. Facial recognition by the brain is made more difficult by heavy makeup, especially if the face was first seen without makeup [73]. Makeup can be so deceptive that it impairs automated facial recognition software [74]. 

As noted before, the application of makeup overcomes the influence of fluctuating estrogen levels on perceptions of attractiveness [27]. By darkening hair color and lightening skin complexion, cosmetics are used to counter this sign of aging by enhancing contrast [62]. Cosmetics also even out skin tone, and for people shown pictures of both made-up and no makeup faces, the number of eye fixations and dwell time were positively correlated with skin color homogeneity [66]. Overall, makeup reduces the perceived age of women, and the older the woman, the greater the reduction of the perceived age [75]. This has significant social benefits, since the use of makeup leads to an increase in the perception of likability, competence and trustworthiness [76], as well as dominance and prestige [77].

Eye makeup increases the appearance of the size of women’s eyes [78] and has the greatest effect on attractiveness as judged by both men and women [65]. Observers look at the eyes of women with eye makeup 40% longer than women without it, and if the rest of the face is made up, the attention to the eyes increased 80% [79]. Lipstick, which increases the appearance of the size and accentuates the shape of the lip, increases the time people spend looking at the lips by 26% [68]. 

Deception by cosmetics has measurable economic consequences. Male patrons at a French restaurant gave tips more often to waitresses who wore makeup and, when they did tip, they gave them larger amounts of money than to waitresses without makeup. There was no difference for female patrons, even though both male and female patrons thought that the waitresses were more attractive when they wore makeup [80]. In macroeconomic theory, the Lipstick Effect (first formulated by cosmetics magnate Leonard Lauder) is the increase in sales of lipsticks during economic downturns, as women turn to small pleasures to compensate for losses and increase their appearance advantage in a more competitive environment. A review of recessions over the past 50 years confirmed that during downturns women tend to increase their purchase of products that enhance their appearance while decreasing their purchase of non-appearance-enhancing products [81].

#### 5.2.2. Deception Detection

Maintaining the value of Costly Signals against deception requires that people are able to detect and punish cheating. Within the brain, the frontal lobe and amygdala are key components of the “lie detector”, as demonstrated by a patient with brain damage in this region who was unable to detect cheating, even though otherwise cognitively normal [82]. The brain is especially attuned to negative information, emphasizing the importance of detecting fakes and posers. Test subjects were significantly more likely to retain and consciously process a human face if it was associated with negative gossip rather than positive or neutral gossip [83]. Since women have the greater stake in detecting who is or is not faking Costly Signals among potential mates and rivals, it is not surprising that many studies have shown that women outperform men in detecting lies and inferring emotions from subtle cues [71].

### 5.3. Self-Deception and Self-Signaling

The best way to convince others of a lie is to believe it yourself, and natural selection is strong enough to build a genetically controlled self-deception mechanism [84]. Many studies suggest that people are objectively accurate in evaluating others but view themselves with an optimism bias.

#### 5.3.1. Self-Assessment

People agree with a correlation of 0.79 about the attractiveness of others, but correlations between self-ratings and objective measures of individual attractiveness are remarkably low: 0.24 for men and 0.25 for women [85]. People maintain an image of themselves that is much better than others perceive. For example, photographs of volunteers were morphed to make them progressively more attractive or less attractive. The subjects, both Western Europeans and Asians, were then invited to pick out from the array of pictures the one that was the accurate representation of themselves. The median choice was a picture that was 20% more attractive [86]. They were also quicker to recognize the more attractive photo than their actual photo, a revealing result considering that people recognize objects more quickly when they match their mental representations.

#### 5.3.2. Self-Deception

The brain can hold a truthful and false belief at the same time because it is composed of domain-specific cognitive modules, each of which evolved for solving a specific problem [87]. For example, the medial prefrontal cortex (mPFC) is integral to processing self-related information but remains quiet when the brain considers non-self-referential information [88]. These modules are not tightly connected and not all are accessible to consciousness. Often, these modules reach a decision which is only later rationalized by conscious thought [89].

Facial attractiveness is so salient to people that it can influence unrelated opinions, such as the perception of fairness. In the Ultimatum Game, a player is offered an arbitrary split of money by a proposer, which he can either accept or reject as unfair. Male players accepted offers as fair from attractive women that they rejected when proposed by unattractive women [90], suggesting that a man can deceive himself into believing an offer is fair just because the proposer is beautiful.

#### 5.3.3. Self-Signaling 

Since the world is a competitive place, people are concerned with their own level of reproductive fitness and social status. However, they cannot know their own status with any certainty, in part because they lie to themselves about their own value. Therefore, they use various signals to themselves to demonstrate their own value [91], such as overcoming challenges or acquiring costly goods or making generous donations, even when no one else knows about it. Self-signaling explains many behaviors of people in secret, private or anonymous purchasing or charity transactions [92]. Many efforts by people to alter or improve their appearance beyond what is conventional or apparent to others, such as some types of cosmetic surgery and even piercings or tattooing on body sites covered by hair or clothing, may be understood as a signal to oneself.

## 6. Conclusions

Dr. Theodosius Dobzhansky, the famous geneticist, wrote in 1973 that “*nothing in biology makes sense except in the light of evolution*”. Since the human form and the brain co-evolved in the 5 million years since our last common ancestor with the apes, we can say that *nothing in beauty makes sense except in light of the brain.* Natural selection favors individuals with greater reproductive fitness and also those who display signs of greater fitness, as well as those who can detect them.

The Costly Signals of fitness for humans include health, youth, and ideal proportions. The brain has evolved modules to perceive facial and body shapes, interpret their meaning, and then assign value—beautiful and attractive, or not. Since men and women have different reproductive strategies, and different sex hormones shape their bodies, the brain is tuned to those features driven by these gender-specific development patterns in reaching decisions about attractiveness. A key finding is that men and women of all cultures agree on which men and women are attractive and who are not.

People use cosmetics and surgical procedures to fake Costly Signals, while at the same time they are always on the lookout to detect cheaters. Deceivers are more convincing when they believe the lie themselves, and we have ample evidence of self-deception in beauty. In this fog of competition, people use self-signaling to indicate to themselves their own worth. None of this makes sense except in light of the neuroscience of beauty.

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
