# Peer review of "Perception and Deception: Human Beauty and the Brain"

_behavsci, 2019, doi:10.3390/bs9040034_

Reviewer 1 Report

This is a concise and mostly well-written review of the evolutionary and neural bases of beauty perception.

The most interesting parts are those involving self-deception and self-signaling (e.g. people being confused as to the attractiveness of their own faces). The paper would benefit from these being greatly expanded.

The least compelling parts are those discussing neural modules and hard-wired mechanisms. Neither universality nor localization of function entails innate modularity. Although these lines of evidence do suggest that some degree of genetic shaping might have been possible, they are much less strongly supporting of this claim than many believe them to be. As such, the paper's current tone is over-confident with respect to the links between the evidence provided and the conclusions drawn.

This paper would represent a better contribution to the literature by a) acknowledging alternative explanations and uncertainty, and b) focusing on the more speculative parts of the paper (while acknowledging the uncertainty of these models).

Author Response

Reviewer 1

Comment 1: This is a concise and mostly well-written review of the evolutionary and neural bases of beauty perception.  The most interesting parts are those involving self-deception and self-signaling (e.g. people being confused as to the attractiveness of their own faces). The paper would benefit from these being greatly expanded.

Response 1:  I appreciate the comments of the reviewer.  The subject of self-deception and self-signaling is new and I have cited the best references, including two books with extensive references.  To go into greater detail might make the review heavily weighted toward this speculative subject.  I think it is best to introduce these concepts here and cover the subject in detail in another.  However, if the editor wishes an expansion of these sections, I will add to the text.

Comment 2: The least compelling parts are those discussing neural modules and hard-wired mechanisms. Neither universality nor localization of function entails innate modularity. Although these lines of evidence do suggest that some degree of genetic shaping might have been possible, they are much less strongly supporting of this claim than many believe them to be. As such, the paper's current tone is over-confident with respect to the links between the evidence provided and the conclusions drawn. 

Response 2: I have added the following to Page 2, Line 83:” While a few loci have been linked together to suggest a pathway for evaluation of beauty, this is not to suggest that this is the only way the brain makes these judgements, and moreover under special conditions the plasticity of the brain may invoke other regions to participate in reaching assessments.”

Comment 3: This paper would represent a better contribution to the literature by a) acknowledging alternative explanations and uncertainty, and b) focusing on the more speculative parts of the paper (while acknowledging the uncertainty of these models).

Response 3: I have added a comment on uncertainty in Response 2.  I am hesitant to put more speculation in this review, which should cover the accepted or largely supported facts.

Reviewer 2 Report

The review is well written and informative, although some passages seem hurried. I believe that the review would benefit from few relevant integrations. 

L13 "The brain employs AT LEAST three modules."

When speaking of the brain, it is alway best not to be too deterministic. I'm sure that the author agrees.

About section 2 “The Evolutionary Biology of Beauty” it’d be informative for the reader to refer back to some specific readings.

L 80 “less research has been reported on body judgments” 

L 225 “Body shape is also a signal of reproductive fitness. People universally have a preference for  shape as expressed in the ratio of waist to hips for men of 0.9, and waist to hips for women of 0.7 [40].”

A most relevant work in this respect is by Di Dio, C, Canessa, N., Cappa, S.F., & Rizzolatti, G. (2011). Specificity of aesthetic experience for artworks: an fMRI study. Frontiers in Human Neuroscience, 5, Article 139. DOI: 10.3389/fnhum.2011.00139 

This work shows common neural patterns associated to attraction to canonical human body shapes and it specifically address the issue of aesthetics using human bodies as stimuli.

L 82 "The most extensive research on the brain regions used in assessing beauty has been reported for  facial recognition [7] and less research has been reported on body judgments [8].” Brain loci used to judge the beauty of faces are distinct in distribution and activation intensity from those used to assess 81 the beauty of non-facial visual art [9], reflecting the evolutionary salience of facial beauty.”

Again, there is critical research on the matter that should be included in the review. In the comparison between facial beauty and body, relevant contributions are by Savazzi, F., Massaro, D., Di Dio, C., Gallese, V., Gilli, G., & Marchetti, A. (2014). Exploring responses to art in adolescence: a behavioral and eye-tracking study. PloS one9(7), e102888;

A most recent brain imaging study is by Di Dio, C, Ardizzi, M, Massaro, D, Di Cesare, G, Gilli, G, Marchetti, A, & Gallese, V (2016). Human, Nature, Dynamism: The Effects of Content and Movement Perception on Brain Activations during the Aesthetic Judgment of Representational Paintings. Frontiers in Human Neuroscience, 9, article 705. 

L83 The brain uses AT LEAST three modules

L84 see also O’Doherty, J., Winston, J., Critchley, H., Perrett, D., Burt, D. M., & Dolan, R. J. (2003). Beauty in a smile: the role of medial orbitofrontal cortex in facial attractiveness. Neuropsychologia, 41(2), 147-155

L227 In one study, men were showN

L231 Ref 59 is to be read and referred to as:

Di Dio, C., Berchio, C., Massaro, D., Lombardi, E., Gilli, G., Marchetti, A. Body aesthetic preference in preschoolers and attraction to canon violation: an exploratory study. Psychol. Rep. 2018 121, 1053-1071.  

L234 Again, a critical work in this respect and – most importantly - with specific reference to aesthetic judgment is by Di Dio, Ardizzi and colleagues (2016 - see above) showing substantial differences in post temp lobe for the aesthetic judgment of paintings representing a human figure (and particularly dynamic subjects) vs painting representing natural environments.

There is a whole new line of research that discusses the possible association between the motor system and aesthetic perception, as also indicated above. Most recent relevant studies in this respect are by Ardizzi, M., Ferroni, F., Siri, F., Umiltà, M. A., Cotti, A., Calbi, M., ... & Gallese, V. (2018). Beholders’ sensorimotor engagement enhances aesthetic rating of pictorial facial expressions of pain. Psychological research, 1-10(see also, Gernot, G., Pelowski, M., & Leder, H. (2017). Empathy, Einfühlung, and aesthetic experience: The effect of emotion contagion on appreciation of representational and abstract art using fEMG and SCR. Cognitive Processing19(2), 147-165.); and also

Cattaneo, Z., Lega, C., Gardelli, C., Merabet, L. B., Cela-Conde, C. J., & Nadal, M. (2014). The role of prefrontal and parietal cortices in esthetic appreciation of representational and abstract art: a TMS study. Neuroimage, 99, 443-450.

L 270 “The next generation produces one sex that favor the Costly Signal and the other sex that displays it” 

Please clarify this concept. It is not very clear, at least to me.

Finally, there are at least two further issues I would hear the author’s opinion about:

- Brain plasticity and cultural/top-down influence. The brain is plastic and subject to context-related perceptual modifications (strong media influence, for instance; or, for example, in some African and Asian cultures, practices (like the use of multiple neck rings) to confer a tribal identity on women that becomes associated with attributes of ideal beauty and wealth. These aesthetic cannons reflect a deviation from the classical conception of beauty. 

- The scope of beauty. With respect to cultural media-driven effects on beauty perception (e.g., supermodel-like women features), how deeply does the author thinks that they actually affect the aesthetic sense? Can that type of beauty be considered deception? How reliable or stable would that effect eventually be when the purpose of a relationship is mating for reproduction?

Author Response

Reviewer 2

The review is well written and informative, although some passages seem hurried. I believe that the review would benefit from few relevant integrations.

Comment 4: L13 "The brain employs AT LEAST three modules."

When speaking of the brain, it is alway best not to be too deterministic. I'm sure that the author agrees.

Response 4: I have added this qualifying phrase

Comment 5: About section 2 “The Evolutionary Biology of Beauty” it’d be informative for the reader to refer back to some specific readings.

Response 5:  These two introductory paragraphs recite classical principles of evolutionary biology that serve as a groundwork.  It would overburden a specialized review to cite textbooks in the underlying science.

Comment 6: L 80 “less research has been reported on body judgments” L 225 “Body shape is also a signal of reproductive fitness. People universally have a preference for shape as expressed in the ratio of waist to hips for men of 0.9, and waist to hips for women of 0.7 [40].” A most relevant work in this respect is by Di Dio, C, Canessa, N., Cappa, S.F., & Rizzolatti, G. (2011). Specificity of aesthetic experience for artworks: an fMRI study. Frontiers in Human Neuroscience, 5, Article 139. DOI: 10.3389/fnhum.2011.00139 

This work shows common neural patterns associated to attraction to canonical human body shapes and it specifically address the issue of aesthetics using human bodies as stimuli.

Response 6: I have added “Similar regions of the brain are used in evaluating sculptures and similarly posed real human bodies.” and the Di Dio reference.

Comment 7: L 82 "The most extensive research on the brain regions used in assessing beauty has been reported for  facial recognition [7] and less research has been reported on body judgments [8].” Brain loci used to judge the beauty of faces are distinct in distribution and activation intensity from those used to assess 81 the beauty of non-facial visual art [9], reflecting the evolutionary salience of facial beauty.”

Again, there is critical research on the matter that should be included in the review. In the comparison between facial beauty and body, relevant contributions are by Savazzi, F., Massaro, D., Di Dio, C., Gallese, V., Gilli, G., & Marchetti, A. (2014). Exploring responses to art in adolescence: a behavioral and eye-tracking study. PloS one9(7), e102888;  A most recent brain imaging study is by Di Dio, C, Ardizzi, M, Massaro, D, Di Cesare, G, Gilli, G, Marchetti, A, & Gallese, V (2016). Human, Nature, Dynamism: The Effects of Content and Movement Perception on Brain Activations during the Aesthetic Judgment of Representational Paintings. Frontiers in Human Neuroscience, 9, article 705.

Response 7:  While these references are in the general topic, they relate to perception of art and representations, while this review is focused on perception of real faces. I have not added these references.

Comment 8: L83 The brain uses AT LEAST three modules

Response 8:  This change has been made

Comment 9: L84 see also O’Doherty, J., Winston, J., Critchley, H., Perrett, D., Burt, D. M., & Dolan, R. J. (2003). Beauty in a smile: the role of medial orbitofrontal cortex in facial attractiveness. Neuropsychologia, 41(2), 147-155

Response 9: While this reference is related, it is focused more on facial expressions (smiling) rather than evaluation of beauty, which is the focus of this review.  I have not added this reference.

Comment 10: L227 In one study, men were shown

Response 10:  I have corrected this typo.  Thank you.

Comment 11: L231 Ref 59 is to be read and referred to as: Di Dio, C., Berchio, C., Massaro, D., Lombardi, E., Gilli, G., Marchetti, A. Body aesthetic preference in preschoolers and attraction to canon violation: an exploratory study. Psychol. Rep. 2018 121, 1053-1071. 

Response 11:  I have corrected the citation

Comment 12: Again, a critical work in this respect and – most importantly - with specific reference to aesthetic judgment is by Di Dio, Ardizzi and colleagues (2016 - see above) showing substantial differences in post temp lobe for the aesthetic judgment of paintings representing a human figure (and particularly dynamic subjects) vs painting representing natural environments.  There is a whole new line of research that discusses the possible association between the motor system and aesthetic perception, as also indicated above. Most recent relevant studies in this respect are by Ardizzi, M., Ferroni, F., Siri, F., Umiltà, M. A., Cotti, A., Calbi, M., ... & Gallese, V. (2018). Beholders’ sensorimotor engagement enhances aesthetic rating of pictorial facial expressions of pain. Psychological research, 1-10(see also, Gernot, G., Pelowski, M., & Leder, H. (2017). Empathy, Einfühlung, and aesthetic experience: The effect of emotion contagion on appreciation of representational and abstract art using fEMG and SCR. Cognitive Processing19(2), 147-165.); and also

Cattaneo, Z., Lega, C., Gardelli, C., Merabet, L. B., Cela-Conde, C. J., & Nadal, M. (2014). The role of prefrontal and parietal cortices in esthetic appreciation of representational and abstract art: a TMS study. Neuroimage, 99, 443-450.

Response 12:  The reviewer points out important and interesting references to judging aesthetic beauty.  However this opens a very large topic of art appreciation, body and facial expressions, and is beyond the scope of this review of human beauty perception.  I have not expanded into this material.

Comment 13: L 270 “The next generation produces one sex that favor the Costly Signal and the other sex that displays it” Please clarify this concept. It is not very clear, at least to me.

Response 13: I have added the explanation: “This is called the Green Beard Effect, after the hypothetical example of two sets of genes that co-evolve, one set that produces a green beard in males and another set that prefers a green beard in females.”

Comment 14:  Finally, there are at least two further issues I would hear the author’s opinion about:

- Brain plasticity and cultural/top-down influence. The brain is plastic and subject to context-related perceptual modifications (strong media influence, for instance; or, for example, in some African and Asian cultures, practices (like the use of multiple neck rings) to confer a tribal identity on women that becomes associated with attributes of ideal beauty and wealth. These aesthetic cannons reflect a deviation from the classical conception of beauty.

Response 14:  The reviewer points out a very interesting area for discussion elsewhere. This review is focused on genetically driven (largely non-cultural) characteristics of beauty that are stable over time.  The reviewer is asking about cultural norms of beauty that change over generations.  These cultural practices serve other evolutionary selection pressures, such as group identity, rather than attractiveness and beauty for purposes of judging reproductive fitness.

Comment 15: - The scope of beauty. With respect to cultural media-driven effects on beauty perception (e.g., supermodel-like women features), how deeply does the author thinks that they actually affect the aesthetic sense? Can that type of beauty be considered deception? How reliable or stable would that effect eventually be when the purpose of a relationship is mating for reproduction?

Response 15: The reviewer raises the interesting point that not all displays of beauty is for judging reproductive fitness.  In some cases, the display of beauty is an exaggeration of a feature to call attention for other reasons (excessive thinness in supermodels displaying high priced clothing).  Most often these are culturally specific and change over time – only a few become such preferences that they become stable (e.g. Green Beards are rare).  This consideration brings in cultural selection and is beyond the scope of this review of evolutionary selection of beauty features and brain perception.

Round  2

Reviewer 1 Report

I am satisfied with the changes that have been made.